# RNA-sequencing of single cholangiocyte-derived organoids reveals high organoid-to organoid variability

Kristin Gehling[1,2] , Swati Parekh[1] , Farina Schneider[3], Marcel Kirchner[4], Vangelis Kondylis[3] , Chrysa Nikopoulou[1] , Peter Tessarz[1,2]

Over the last decades, organoids have been established from most of the tissue-resident stem and iPS cells. They hold great promise for our understanding of mammalian organ development, but also for the study of disease or even personalised medicine. In recent years, several reports hinted at intraculture organoid variability, but a systematic analysis of such heterogeneity has not been performed before. Here, we used RNA-seq of individual intrahepatic cholangiocyte organoids to address this question. We find that batch-to-batch variation is very low, whereas passage number has a profound impact on gene expression profiles. On the other hand, there is organoid-to-organoid variability within a culture. Using differential gene expression, we did not identify specific pathways that drive this variability, pointing towards possible effects of the microenvironment within the culture condition. Taken together, our study provides a framework for organoid researchers to properly consider experimental design.

## Introduction

Organoid cultures have been present in modern research laboratories for over a decade and are thought to bridge the gap between 2D and 3D-tissue culture (Broutier et al, 2017; Aizarani et al, 2019; Lancaster & Huch, 2019). Organoids can be derived either from pluripotent cells, such as embryonic or induced pluripotent stem cells, but also from tissue-resident stem and progenitor cells (Prior et al, 2019). In particular, iPSC-derived organoids can give rise to remarkably complex structures (Takebe et al, 2013). Recently, gene regulatory network analysis using CellNet (Cahan et al, 2014) in combination with CRISPRCas based engineering was used to generate complex organoids (Velazquez et al, 2021). Although high complexity as well as disease modelling can be nowadays derived in iPSC-derived organoids, they lack epigenetic information of the tissue of origin, which might hamper analysis of complex states,

such as cancer. Thus, besides complex multilineage organoids, 3D structures have been derived from tissue-resident progenitors (Broutier et al, 2017).

In the case of the liver, organoids are based on hepatocytes and cholangiocytes. Hepatocyte-derived organoids, so-called "Hep-Orgs," consist mostly of progenitors and hepatocytes (Hu et al, 2018). In contrast, "Chol-Orgs" are derived from EPCAM+ or Lgr5+ biliary epithelial cells (also referred to as intrahepatic cholangiocyte organoid, ICO [Marsee et al, 2021]) and have the potential to differentiate into either hepatocytes or cholangiocytes (Huch et al, 2013). Upon in vitro differentiation, cholangiocyte-derived organoids will give rise to functional cells that display hepatocyte characteristics, such as increased glycogen storage, LDL uptake or albumin secretion. The differentiated organoids can be successfully transplanted into animal models of liver disease, where they acquire mature hepatocyte characteristics and ameliorate disease phenotype (Huch et al, 2013; Broutier et al, 2016). This murine model system is frequently used in liver biology and allows repopulation with hepatocytes.

In recent years, intense effort was put on the establishment of 3D cultures from various organs and nowadays, organoids can be grown representing virtually any tissue. The vision in ongoing consortia is to exploit the tissue-like features of organoids to understand the development of human disease (Rajewsky et al, 2020) and thus, similar to an organismal atlas, organoids are also currently profiled to generate an overview of cell types present as part of the human cell atlas project (Bock et al, 2021).

To date, organoids represent the model system, which most closely resembles the tissue of origin. The multicellular nature of organoids makes them a sophisticated but variable model, which displays heterogeneity (Lancaster & Knoblich, 2014) and can strongly depend on many extrinsic factors, such as culture conditions (Criss et al, 2021). However, we still need to better understand the drivers of batch-to-batch and organoid-to-organoid variability within the same culture. To address these questions, we profiled single intrahepatic cholangiocyte organoids from four different batches and passage numbers (Fig 1A). To allow an easier isolation of single organoids, we initially set up shaking cultures

[1]Max Planck Research Group "Chromatin and Ageing," Max Planck Institute for Biology of Ageing, Cologne, Germany   [2]Cologne Excellence Cluster on Cellular Stress Responses in Aging-associated Diseases (CECAD), Cologne, Germany   [3]Institute for Pathology, University Hospital Cologne, Cologne, Germany   [4]FACS and Imaging Core Facility, Max Planck Institute for Biology of Ageing, Cologne, Germany

Correspondence: cnikopoulou@age.mpg.de; ptessarz@age.mpg.de

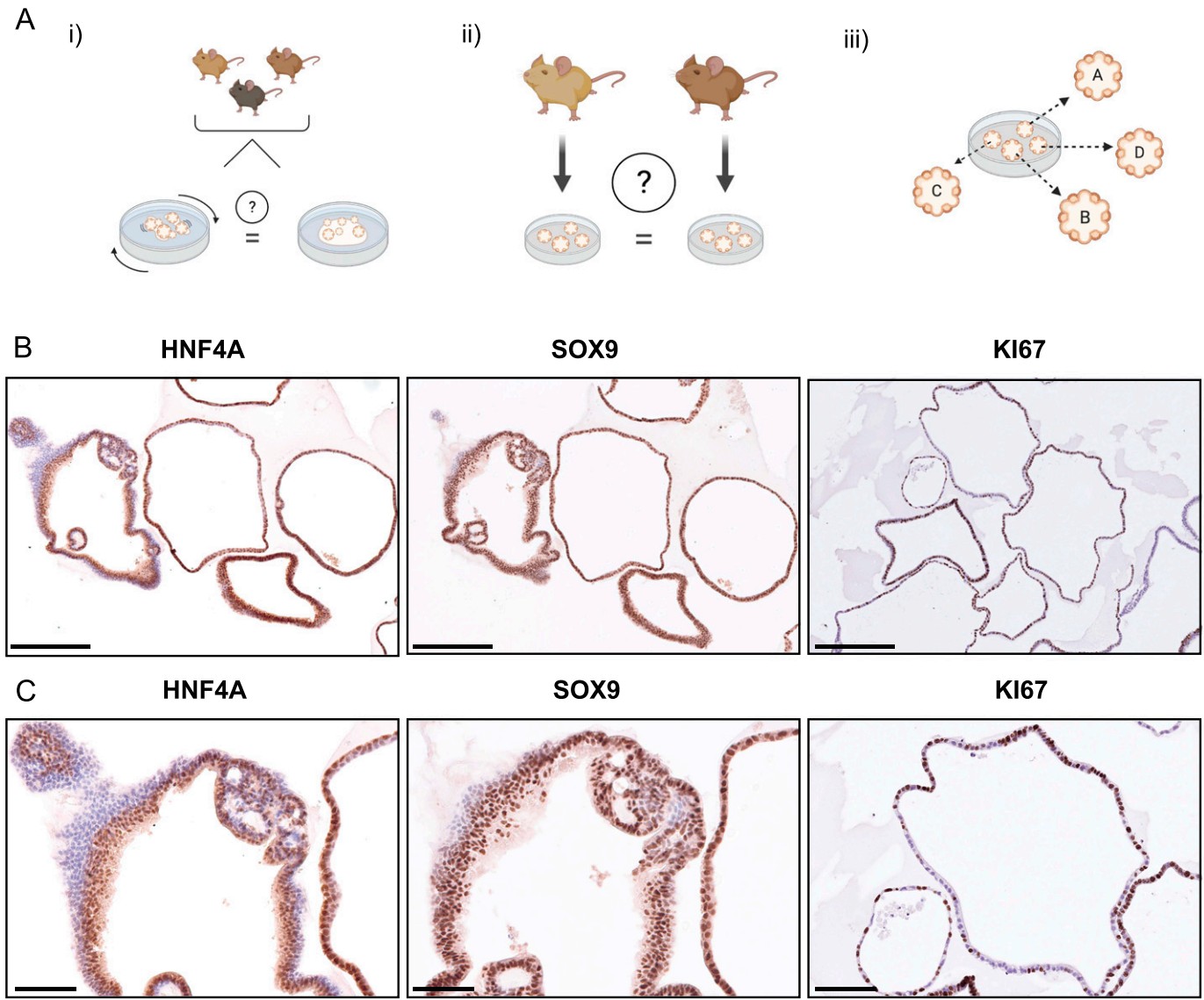

**Figure 1. Rationale and setup of the study.**
**(A)** This study aims to investigate organoids (i) grown in a shaking culture and evaluate them as an alternative to dome culture, (ii) generated from different animals to interrogate batch effects, and (iii) analysis of single organoids to assess the heterogeneity within a culture. **(B)** Immunohistochemistry of the same sample of intrahepatic cholangiocyte organoids (of 3-mo-old male mice) for HNF4, SOX9, and Ki67. Scale bar: 250 $\mu$m. **(B, C)** Magnified areas of immunostained organoids in (B). Scale bar: 100 $\mu$m.

from organoids derived from adult livers and compared their gene expression programs with those of the same organoids growing in domes. Gene expression profiling revealed a striking change in the transcriptional program towards a more progenitor-like state, with an increase in proliferation-related terms. Next, we isolated single, intact organoids, which were macroscopically evaluated before RNA extraction and library preparation. This approach resulted in the generation of 35 single organoid libraries from four organoid batches. The batch-to-batch variation was very low, even between batches generated by different scientists. However, the variability between organoids within a given batch was much larger, but was not determined by size or overall cell cycle state. RNA-seq of organoids from the same batch at early and late passages

(p4 versus p11) demonstrated the impact of passage number on gene expression differences. Taken together, we provide a resource that addresses confounding factors in organoid culture, which will hopefully help the community with their experimental design.

## Results

### Heterogeneity within an organoid culture

To address overall heterogeneity within organoid dome cultures, we initially performed several stainings for progenitor or proliferation markers. We observed a considerably variable amount of

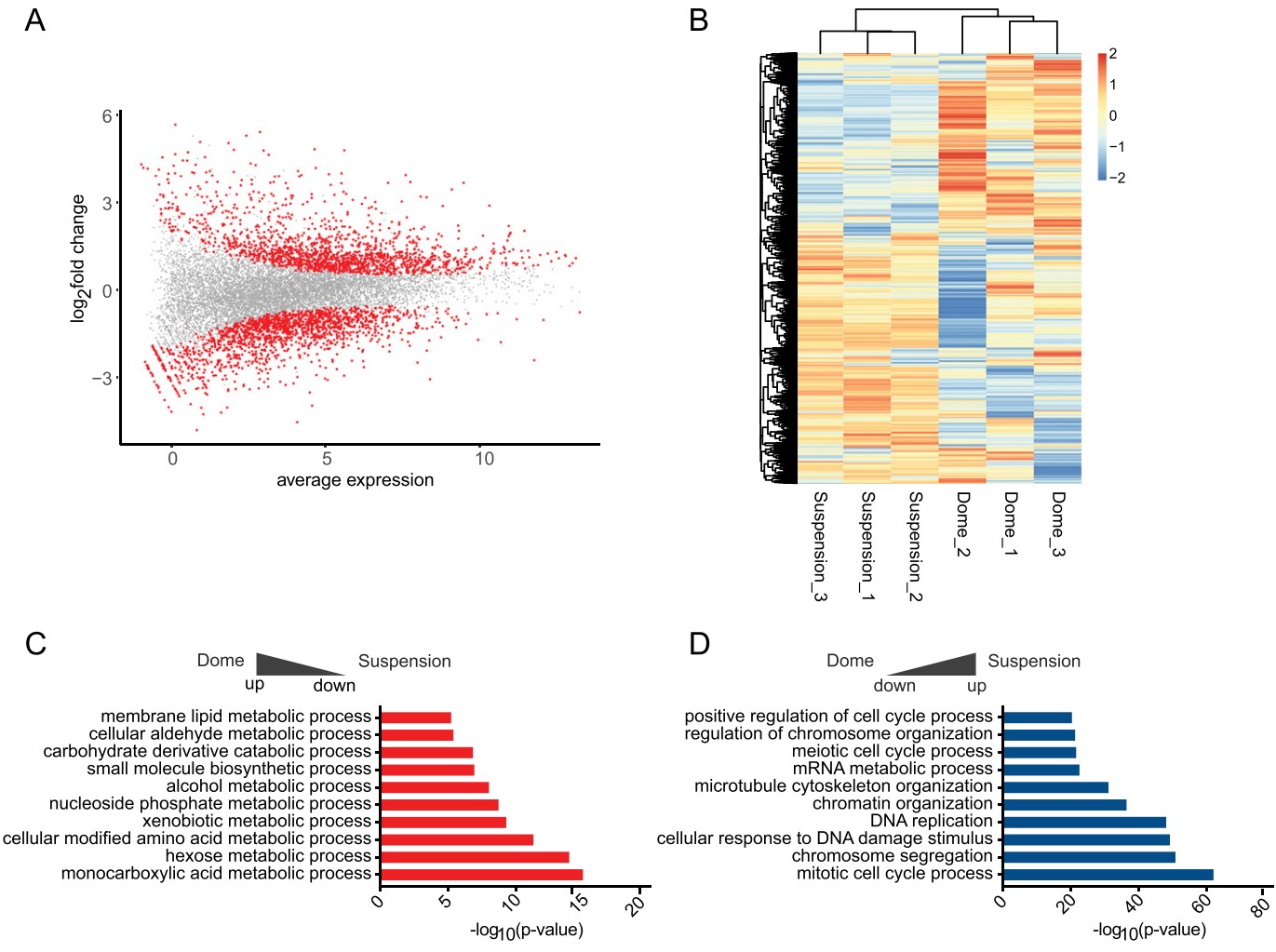

**Figure 2. Differential gene expression analysis of dome and suspension-cultured ductal-derived organoids.**
**(A)** MA-plot showing the log$_2$ ratio and average expression values of differentially expressed genes (DEGs). Significant genes after *P*-value adjustment are displayed in red, n = 3; biological replicates. **(B)** The heat map shows the normalised expression values generated by limma-voom for the fraction of significant DEGs. Heat map data are clustered by both row and column, and scaled by row. **(C, D)** Metascape enrichment analysis of DEGs comparing dome and suspension culture. **(C, D)** Terms up-regulated in dome are shown in (C), whereas terms up-regulated in suspension-grown cells are shown in (D).

Ki67 positive cells between organoids, as well as differences in the morphology of the organoids and slight differences of the SOX9 and HNF4A staining, which suggests that there can be a high degree of variability between individual organoids (Figs 1B and C and S1). These results prompted us to investigate the heterogeneity between organoids further to understand if there are specific pathways that might explain the observed variability.

## Comparison of dome versus shaking culture

To enable simple isolation of individual organoids, we wanted to set up culture conditions, in which cells would grow in a low percentage of Matrigel that would be amenable to pipetting individual organoids. Culture conditions using low Matrigel concentrations have been reported before to generate a large number of cells (Schneeberger et al, 2020). To compare the low-percentage Matrigel cultures with the classic cultures of organoids in domes, we either

maintained organoids in domes or moved them to shaking cultures (see the Materials and Methods section for details) and collected RNA to perform RNA-seq. We assessed the gene expression profile for both conditions using 3′-end RNA-seq, which was subsequently analysed using zUMIs (Parekh et al, 2018).

After filtering for an adjusted *P*-value below 0.05, 3,225 genes were found differentially expressed between dome and suspension cultures, of which 1,635 were down and 1,590 up-regulated (Fig 2A and Table S1). The heat map of differentially expressed genes showed a clear separation between the two conditions and suggested higher variability between samples in the dome cultures (Fig 2B). To investigate possible pathways and functions behind differentially expressed genes, we performed gene ontology enrichment analysis using Metascape on genes with an absolute log$_2$ fold-change >1, which resulted in 1,015 up- and 1,172 down-regulated genes (Zhou et al, 2019). Most of the terms enriched in dome-cultured organoids were associated with metabolic terms (genes included

Cyp2b10, Cyp2c29, Cyp2j6; Fig 2C), whereas suspension cultures showed enrichment in terms that were connected to proliferation, such as "mitotic cell cycle process" or "DNA replication" (Cdk1, Cdk11b, Cdc45, and Cdk4; Fig 2D and Table S1). Taken together, the observed gene expression differences suggested that organoids grown within a Matrigel dome represented a more differentiated (hepatocyte-like) state than those within the shaking cultures, which was dominated by proliferation-associated terms.

## Passage number is the strongest contributor to gene expression changes in organoid cultures

To assess the heterogeneity of organoids, new cultures were established from four 3-mo-old male mice. Each culture originated from the liver of one animal. After establishing organoid cultures in domes, organoids were broken down into fragments, which were seeded in suspension culture and maintained there for at least two passages (see also the Materials and Methods section). Individual organoids from different wells of the same biological replicate were carefully transferred into a new 24-well plate with a pipette. Only non-apoptotic organoids, without a dark necrotic lumen, were selected, and a variety of sizes was included. Representative organoids are displayed in Fig 3A. The images were used to calculate the 2D surface area (Fig S2A and B). Organoids from Set 1 were analysed after passage 11 and displayed a similar morphology within the set regarding the evenness of the lumen compared with all other sets. Sets 2 and 3 were harvested at passage three and Set 4 at passage four. In total, we generated 42 RNA-seq libraries across the four sets, of which 35 libraries passed quality control and filtering for a minimum library size of four million reads. To visualise the variance within the dataset, principal component analysis (PCA) was performed (Fig 3B and C). The library size of each sample is annotated in counts per million and represented by the size of the dot. Importantly, library size differences (though present) did not drive separation between the individual organoids. Interestingly, whereas organoids within sets 2–4 clustered closely together, Set 1 formed a distinct cluster (Fig 3B and C). Initially, passage number was not considered when setting up this experiment, but the analysis pointed towards passage as a major separator between Set 1 and sets 2–4. This indicated that changes occurred in these organoids upon prolonged culturing. To investigate this potential passage effect further, we performed differential expression testing between all sets. The analysis was performed using the linear models approach within limma (Law et al, 2014; Ritchie et al, 2015). As this study did not involve a simple control-treatment design, the four different sets were juxtaposed within six comparisons. The expression values of significantly expressed genes (adj. P-value < 0.05) were subsequently displayed in a heat map (Fig 3D and Table S2) and data were clustered by sample (i.e., column). The heat map replicated the trend seen in the dimensionality reduction analysis above and showed most of the Set 1 to be clustering in a separate branch. In contrast, the other three sets appeared ordered according to their passage number, suggesting that indeed, gene expression was dominated by passage number. To address the changes in gene expression on a more functional level, we first compared the number of

differentially expressed genes across the sets (Fig S3). Not surprisingly, comparisons with Set 1 showed the strongest differential expression. We then chose the intersection between these comparisons with Set 1 and performed GO term enrichment analysis for up-regulated genes in Set 1 (Fig 3E). The enrichment map of GO terms pointed towards a strong enrichment for hepatocyte-specific functions, indicating that prolonged culture leads to a more mature hepatocyte phenotype. The enrichment for a more progenitor-like state is evident in the GO analysis of down-regulated genes (Fig 3F), in which proliferation and developmental terms were dominating.

### Impact of passage number on gene expression

The observation that prolonged passage number changed gene expression profiles of organoids prompted us to explore this more systematically. To do so, we performed RNA-seq on the same sets of organoids, once at passage 4 and then again at passage 11. PCA analysis was able to clearly separate the two passages. Interestingly, organoids of passage 4 clustered closely together (Fig 4A), supporting the notion that generally batch-to-batch variability was low, whereas organoids in passage 11 showed more variability. These data indicated that continued passaging of organoids changed gene expression programs and increased variability compared with early passage organoids. To further explore this dataset, we performed differential gene expression analysis (Fig 4B). 424 genes were significantly down-regulated and 825 genes up-regulated in passage 11 in comparison with passage 4 organoids (Fig 4C and Table S3). We then performed gene ontology analysis of the up- and down-regulated genes (absolute $\log_2$FC > 1). Terms associated with down-regulated genes were associated with proliferation and cell cycle progression (Fig 4D), whereas up-regulated terms were related to interferon and TNF signalling, cell motility and substrate adhesion, as well as general stress response pathways (Fig 4E), indicating that organoids are functionally distinct between the two different passage numbers.

### Organoid-to-organoid variability

We assessed the heterogeneity within each set using PCA (Fig 5A–D), which indicated a high degree of variability between individual organoids. To understand in more detail what drove the separation, we analysed the genes driving the separation of PC1 and PC2 for each set (Table S4). However, we did not identify any enriched term for the differentially expressed genes, indicating that there were no changes in specific pathways between the individual organoids. As the unbiased approach did not yield any specific pathway, we took a candidate-based approach. We chose marker genes that report proliferation status, response to Wnt signalling (an essential contributor to long-term expansion [Huch et al, 2015]) or differentiated versus progenitor-like state (Fig 5E). The heat map confirms the high level of heterogeneity between the individual organoids. It also corroborates the finding that organoids from Set 1 have a more hepatic-like state compared with the others. Interestingly, even within a given culture, such as Set 2, organoids can express high levels of mature hepatic markers (e.g., Fah and Ldlr),

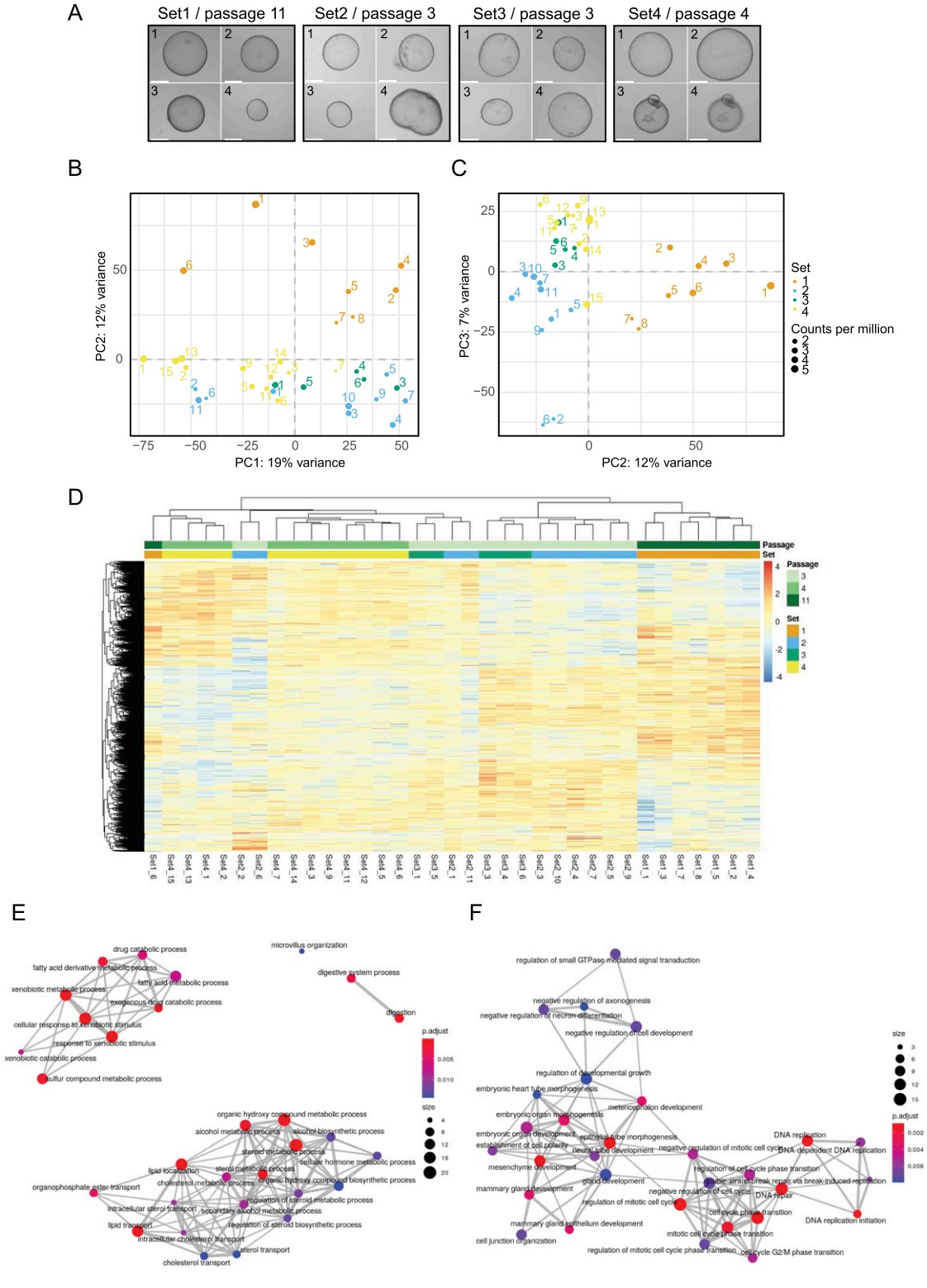

**Figure 3. Analysis of single organoids.**
**(A)** Representative microscopy images taken from individual organoids. The passage number on the day of sorting is indicated for each set. Sample numbers are written in the top left corner of each image. Scale bar: 275 μm. **(B, C)** The grouping of organoids after dimensionality reduction by principal component analysis for the first two PCs and PC 2 and 3. The distribution of variance among the PCs is plotted along, with each proportional contribution to the variance in the axis labels. The dot size indicates library size in counts per million. **(D)** Heat map of expression values after differential expression analysis. Significant genes after *P*-value adjustment are clustered by both row and column, and scaled by row. **(E)** Enrichment map showing GO term analysis of up-regulated DEGs shared against Set 1. **(F)** The intersection of down-regulated DEGs from Set2vs1, Set3vs1, and Set4vs1 plotted as enrichment map.

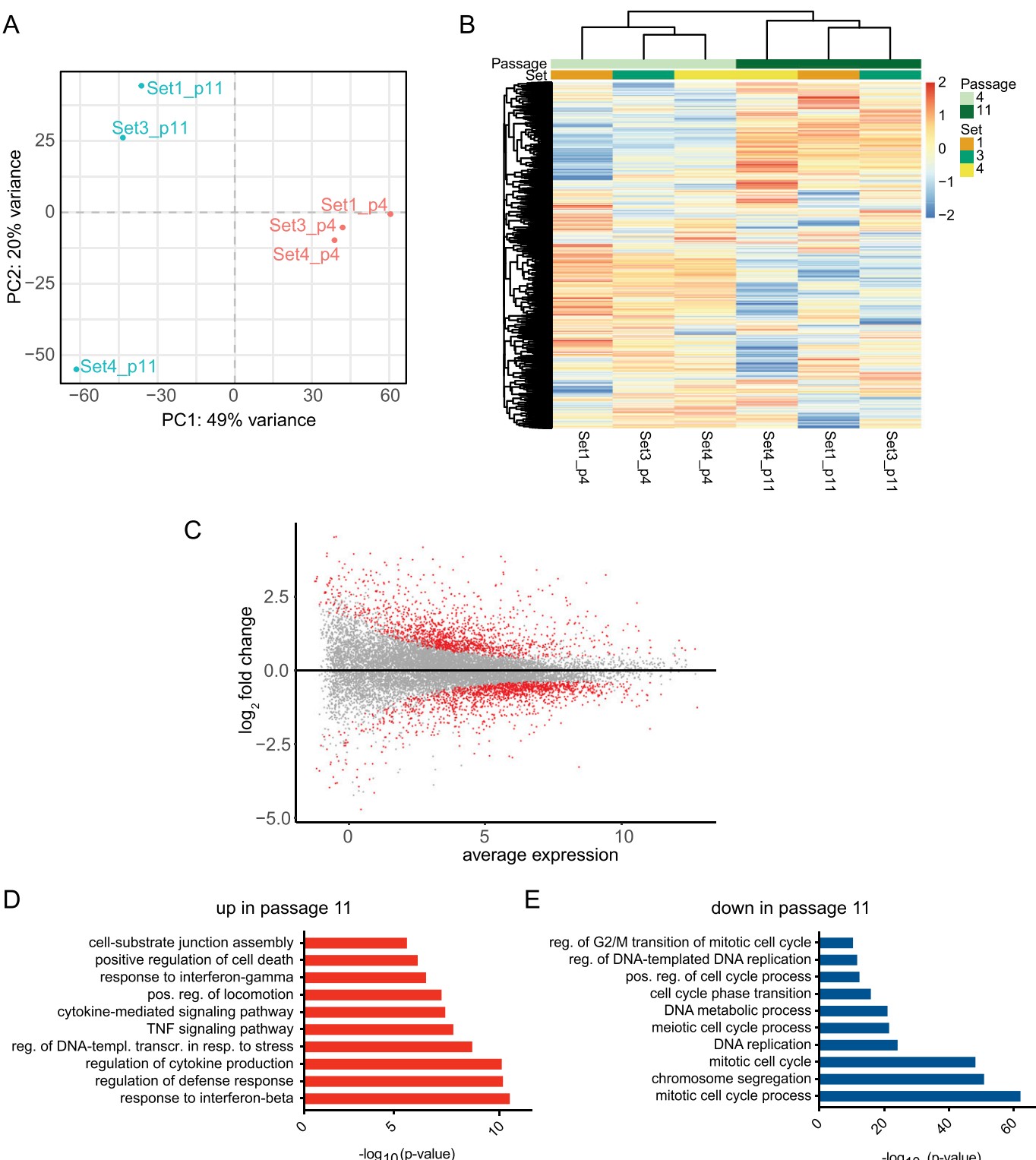

**Figure 4. Impact of passage number on gene expression programs.**
**(A)** Principal component analysis of organoids of the same batch at passage 4 (p4) and passage 11 (p11). Set numbers are given. **(B)** Heat map of expression values after differential expression analysis. Significant genes after *P*-value adjustment are clustered by both row and column, and scaled by row. **(C)** MA-plot showing the $\log_2$ ratio and average expression values of differentially expressed genes (DEGs). Significant genes after *P*-value adjustment are displayed in red, n = 3; biological replicates. **(D, E)** Metascape enrichment analysis of DEGs comparing early and late passage numbers. **(D, E)** Terms up-regulated at passage 11 are shown in (D), whereas terms down-regulated at passage 11 are shown in (E).

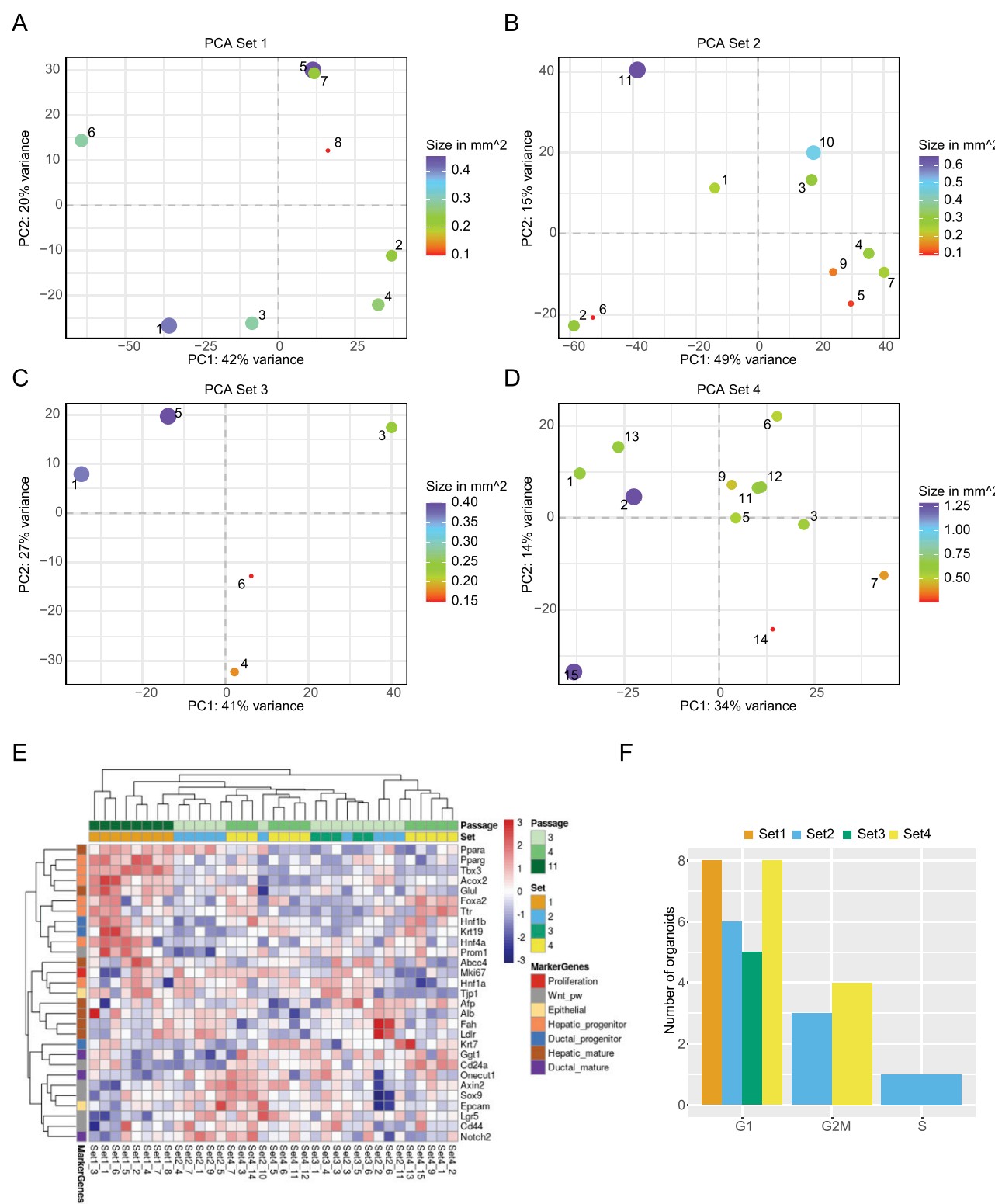

**Figure 5. Inter-organoid heterogeneity.**
**(A, B, C, D)** Projection of organoids within each set after dimensionality reduction by principal component analysis, including the annotation of each organoid's size as dot size. The variance for each PC is indicated in the axis labels. **(E)** Heat map showing the expression values for marker genes after limma-voom normalisation, with a minimum count = 1. Rows are annotated by official gene symbols and colours indicating marker for proliferation (red), Wnt pathway (grey), epithelial (yellow) as well as progenitor and mature cell types for hepatic (orange, brown) and ductal (blue, violet). Data are clustered hierarchically by row and column and scaled by row. **(F)** Bar graph showing number of organoids for each cell cycle phase, coloured by set, after analysis with cyclone.

whereas others were enriched in ductal and progenitor markers (e.g., Notch2, Lgr5, or EPCAM), indicating that organoids within one culture can lean towards two different fates.

To understand whether the size of an organoid might impact gene expression programs, we went back to the PCA analysis, in which the size of the organoid is indicated (Fig 5A–D). However, no clear relationship between transcriptome and size was observable. To confirm this result, we plotted the loadings of the first two principal components against the size of the organoids and fitted a regression line to the data (Fig S4A and B). Although there was a good correlation between size and the PC loadings in some instances, this was not consistent in all cases. In conclusion, organoid size does not seem to be a driver of the individual transcription programs within single organoids. Finally, we evaluated whether the overall proliferative state of organoids might contribute to the differences in transcriptional states. As a proxy for the level of proliferation, we performed a cell cycle analysis using Cyclone (Scialdone et al, 2015). Cyclone is a machine learning–based approach allowing cell cycle stage prediction based on a reference transcriptome. Here, a sample is assigned to G1 or G2M, when it reaches the threshold of 0.5 for the particular phase. If both G1 and G2M scores stay below 0.5, the sample will be categorised as S-phase. Most of the organoids were assigned to the G1 phase (Fig 5F). Although some organoids were predicted to fall more into the G2M phase, this assignment did not correlate with the clustering in the PCA. Still, most organoids showed variability in the cell cycle score. These results were also supported by FACS analysis (Fig S4C and D), showing that most of the cells in the organoid cultures are indeed in the G1 phase. In conclusion, proliferative states of individual organoids will contribute to the organoid-to-organoid variability as well as size, but these two parameters alone were not able to predict the stark differences in gene expression programs.

# Discussion

Organoid-to-organoid variability has been observed and reported before for epithelial organoids (Hof et al, 2021) and is particularly prevalent in organoids recapitulating high tissue complexity, such as brain organoids (Quadrato et al, 2017; Velasco et al, 2019). Recent studies have analysed in-depth the effect of different culture conditions and treatments on gene expression variability (Criss et al, 2021) and the donor batch effect of different donors on cultures of human gut organoids (Mohammadi et al, 2021). Here, we report that less complex organoids, derived from genetically identical mice, also show a high degree of variability from organoid to organoid. However, reproducibility (as measured in bulk assays) was high between several batches of organoids. In this context, we would like to point out that three different researchers generated organoids used in this study. Together, these findings point out that the protocol for the generation of cholangiocyte-derived organoids from liver tissue is very robust.

Despite the high reproducibility on a bulk level, organoid-to-organoid variability was obvious and marker gene analysis suggested a variety of different cellular states among the organoids.

However, the underlying reason for the variability was less clear. Organoid size and proliferation state, as delineated from predicted cell cycle stage, contributed to the overall variability, but their impacts were not large enough to fully explain the extent of this variability. Given the contribution of culture conditions on variability as seen in the shaking organoid culture, it is reasonable to assume that intra-well conditions might be a strong driver of culture variability (Snijder & Pelkmans, 2011). Indeed, during culturing, assemblies of organoids of various sizes and numbers can be observed, as well as single organoids (Fig S5). Thus, cell-to-cell or cell-to-Matrigel contacts will be different in each scenario and might change the underlying transcriptional program. This might ultimately lead to different signalling events as well. Taken together, to grasp biological meaningful signals, scientists need to include multiple technical replicates from organoid cultures of different biological hosts. In addition, depending on the question, specific culture conditions, passaging, and culturing time is an important consideration as it can change the cellular state within the organoids.

# Materials and Methods

### Organoid cultures used in this study

Information about the organoid cultures used in each experiment can be found in Table S5.

### Initiation of intrahepatic cholangiocyte organoid cultures

3-mo-old male C57BL/6N mice (no littermates) fed with standard chow were maintained in the mouse facility of Max Planck Institute for Biology of Ageing and euthanized according to approved ethical guidelines (granted by the Landesamt für Natur, Umwelt und Verbraucherschutz Nordrhein-Westfalen).

For comparing the culture methods, cultures were made by pooling digested material from different animals each to ensure sufficient material (Pool A = 2 mice; Pool B = 4 mice; Pool C = 3 mice, depending on availability at time of dissection). Organoid cultures for the heterogeneity experiment (Set 1–Set 4) were established from the liver of only one mouse for each set. No physiological abnormalities, such as tumours, were observed during dissection of the mice.

Livers were excised postmortem and digested according to the manufacturer's protocol (HepatiCult, StemCell Technology, 06030) with the following modifications. A total of three digestion cycles were needed to dissolve the 3–5 mm pieces of liver tissue. The 70 $\mu$M strainer was omitted during duct isolation, and the pooled supernatant only passed through a 35 $\mu$M cell strainer. Flow-through was discarded, the strainer was reversed onto a falcon tube and 10 ml cold Advanced DMEM/F-12 (12634028; Thermo Fisher Scientific) was added to release the hepatic ducts from the strainer. To ensure the detachment of big fragments, the bottom of the filter was scraped with a P1000 pipette and the remaining fragments were repeatedly collected with a total of 2 ml Advanced DMEM/F-12.

## Culture in Matrigel domes

The pelleted ducts were cultured in 30 $\mu$l Matrigel (Corning Matrigel Growth Factor Reduced Basement Membrane Matrix, Phenol Red-free, LDEV-free, product number 356231, with protein concentration >8–11 mg/ml) domes as described in the Supplementary Protocol for Mouse Hepatic Progenitor Organoid Culture (Cat. no. 06030). Organoid structures arose within 6 d. Organoids were maintained in a 37°C incubator at 5% $CO_2$ and 20% $O_2$. The medium was changed every 2 d and cultures were passaged every 5–7 d with mechanical dissociation of the Matrigel by pipette-mediated shearing. The cultures were tested regularly for mycoplasma contamination using the MycoSPY Master mix (M020025; Biontex).

## Suspension cultures in 10% Matrigel

Organoids were always initiated in dome cultures. To set up organoids in dilute suspension cultures, organoids in Matrigel domes were broken down into fragments and passaged into a suspension culture. 50 $\mu$l of a 1:10 Matrigel/complete HepatiCult mixture was mixed with the fragment pellet and pipetted into one well of a cooled 24- or 12-well plate (83.3922.500 or 83.3921.500; Sarstdedt), already containing 950 $\mu$l of the Matrigel/HepatiCult mixture. The cultures were maintained on an orbital shaker at 80 rpm. in a 37°C incubator, 5% $CO_2$, 20% $O_2$. Every 3–4 d, the organoids were passaged by mechanical breakdown of the Matrigel during pipette-mediated shearing. This protocol follows established protocols (Kumar et al, 2019) and is described by StemCell Technologies as a suggested method to enhance organoid concentration and numbers (https://www.stemcell.com/products/hepaticult-organoid-growth-medium-mouse.html#section-product-use) and was previously also described for liver-derived organoids to enhance the number of Lgr5 cells in culture (Schneeberger et al, 2020).

## Selection of individual organoids

Organoids were maintained as dilute suspension cultures. A sterile and RNAse-free work environment was established by wiping surfaces with 70% EtOH and RNeasy (049912; APPLICHEM). The tip of P200 filter tips was cut with a sterile razor to allow the pipetting of organoids in a volume of 10–20 $\mu$l without disturbing the lumen. Using a Leica M80 Stereo Microscope, individual organoids were carefully transferred into a neighbouring well with DPBS (14190250; Thermo Fisher Scientific) and subsequently added to a cooled 24-well plate with 10 $\mu$l droplets of Advanced DMEM/F-12, resulting in one organoid per well.

## Immunohistochemistry

Organoids were fixed in situ in 4% PFA for 1 h at RT, washed twice with 1X PBS and isolated by mechanical disruption of the Matrigel. The organoids were then processed for paraffin embedding. Sections of paraffin-embedded samples were deparaffinised by immersion of the slides into the following buffers; 20 min in Xylol, 2 min in 100% EtOH, 2 min in 96% EtOH, 2 min in 75% EtOH and washed two times in $H_2O$ for 5 min each. Endogenous peroxidase was quenched by immersion for 15 min in peroxidase blocking buffer (0.04 M Na citrate, pH 6.0, 0.121 M $Na_2HPO_4$, 0.03 M $NaN_3$, and 3% $H_2O_2$). After three washes with tap water, slides were subjected to heat-induced epitope retrieval with 10 mM NaCitrate, 0.05% Tween-20, pH 6.0, washed 5 min with 1× PBS, blocked 60 min with Blocking buffer (1% Albumin, 0.2% Fish Skin Gelatin, 0.1% Triton X-100, and 0.05% Tween-20 in PBS) + 160 $\mu$l/ml AvidinD (no. SP-2001; Vector) and incubated with primary antibodies diluted (1:400 Ki67 [ab15580; Abcam], 1:200 SOX9 [AB5535; Merck], and 1:200 HNF4a [ab41898; Abcam]) in blocking buffer + 160 $\mu$l/ml Biotin (no. SP-2001; Vector) overnight at 4°C. After three 5-min washes with PBS +0.05%TWEEN (PBST), the samples were incubated with the secondary antibody (anti-rabbit biotin, Perkin Elmer NEF813 or anti-mouse biotin, Biozol BA-9200) 1:1,000 diluted in blocking buffer for 1 h at room temperature, followed by three 5 min washes with PBST and incubation for 30 min with 1× PBS containing1:60 AvidinD and 1:60 Biotin (ABC kit, Vector PK6100). After three 5-min washes with PBST, the samples were stained with 1 drop of DAB chromogen in 1 ml substrate buffer (ImmPACT, SK4105; Vector), washed with 1× PBS, and counterstained with hematoxylin for 4 min, washed with tap water, and dehydrated 1 min in each buffer; 75% EtOH, 96% EtOH, and 100% EtOH, xylol and mounted with Entellan.

## Microscopy

Immunohistochemistry images were taken using the slide scanner Hamamatsu S360 and analysed with the NDP.view2 software. Single organoid Images were taken with the EVOS FL Auto 2 Imaging System in standard bright-field with a 4×/0,13 NA or a 10×/0,25 NA objective. To calculate the organoid area, acquired raw files were analysed with an automated macro in FIJI (ImageJ version 2.1.0/1.53c) using the following steps: Gaussian Blur with a radius of $\Sigma$ = 3, Auto Threshold method = MaxEntropy followed by the "Fill Holes" function of the binary mask. Subsequent "Analyze Particles" delivered the desired areas (size of organoids).

Whole-well organoid images were taken with the EVOS FL Auto 2 Imaging System in standard bright-field with a 4×/0,13 NA. Individual images were stitched together to recreate the image of the whole well.

## mRNA and total RNA extraction

Total RNA of bulk organoids was extracted with the Zymo Quick-RNA Microprep kit (R1050; Zymo Research). The mRNA of single organoids was extracted with the Dynabeads mRNA DIRECT Kit by Thermo Fisher Scientific (#61011) with the following modifications and self-made buffers, detailed compositions are found on the manufacturer's website. In brief, for each sample (i.e., single organoid), 10 $\mu$l of resuspended beads were transferred to a 2 ml DNA low-binding tube and placed on a DynaMag-2 magnet stand. The supernatant was discarded, the magnet removed and beads were resuspended in 50 $\mu$l room temperature Lysis/Binding Buffer. With a volume of 150 $\mu$l room temperature Lysis/Binding Buffer, each organoid was transferred to a 1.5-ml low-binding tube already containing the equivalent amount of buffer. The content was pipetted up and down five times to allow lysis. The following steps were performed according to the manufacturer's instructions. The mRNAs were normalised for library preparation input by measuring

actin (Actb Fw: 5′- CAGCTTCTTTGCAGCTCCTT Rv: 5′- CACGATG-GAGGGGAATACAG) expression via quantitative (q)PCR. The Luna Universal Probe One-Step RTqPCRKit by New England Biolabs (#E3005S) was used to combine RT with qPCR. A scaling factor for the mRNA for consecutive library preparation (protocol below) was calculated with the following formula: cqmax-cqmean = 2 scaling factor. Cqmean was calculated from two independent qPCR runs. Cqmax was set to 21. The maximum input volume for RT is 6.4 $\mu l$, thus 6.4 was divided by each sample's scaling factor and yielded a normalised amount of input mRNA.

### RNA-seq

RNA libraries were created as previously described (Allmeroth et al, 2021). In brief, equal amounts of mRNA or total RNA per sample were used for cDNA synthesis with Maxima H Minus reverse transcriptase (EP0751; Thermo Fisher Scientific). During RT, unique barcodes, including unique molecular identifiers (UMIs), were attached to each sample. After cDNA synthesis, all samples were pooled and processed in one single tube. DNA was purified using AmpureXP beads (A63880; Beckman Coulter) and the eluted cDNA was subjected to Exonuclease I treatment (M0293S; New England Biolabs). cDNA was PCRamplified for 12 cycles and subsequently purified. After purification, cDNA was tagmented in 10 technical replicates of 1 ng cDNA each using the Nextera XT Kit (FC-131-1024; Illumina), according to the manufacturer's instructions. The final library was purified and concentration and size were validated by Qubit and High Sensitivity TapeStation D1000 analyses. Paired-end sequencing was performed on Illumina NovaSeq 6000. Fastq files were processed with zUMIs (version 2.9.5) using its Miniconda environment (Parekh et al, 2018) with STAR index 2.7 (Dobin et al, 2013), SAMtools (version 1.9) (Li et al, 2009) and "featureCounts" from Rsubread (version 1.32.4) (Liao et al, 2013). The reads were mapped to *Mus musculus* (mm10) with Ensembl annotation version GRCm38.93. Libraries were down-sampled within zUMIs, depending on library size variability. Downstream computational analysis was conducted in R (version 3.6.3). The count matrix was normalised and filtered with edgeR (version 3.28.1) (Robinson et al, 2010) using "filterByExpr" with the min.count = 3. For differential gene expression analysis, the limma-voom approached by limma (version 3.42.2) (Ritchie et al, 2015) was used with a pipeline including linear model fit (lmFit) and *P*-value adjustment for multiple testing ("topTableF" with adjust.method = "BH," "decideTests" with method = "global"). Obtained sets of genes were further analysed, for example, through gene enrichment analysis with MetaScape (Zhou et al, 2019). Intersections were visualised with UpsetR (version 1.4.0) (Conway et al, 2017), heat maps created with pheatmap, version 1.0.12 (Kolde, 2019) and cell cycle analysis with cyclone (Scialdone et al, 2015). Results were plotted with ggplot2, version 3.3.3 (Wickham, 2011).

### FACS analysis

For the FACS analysis, we used the Phase-Flow FITC BrdU Kit (#370704; BioLegend) according to the manufacturer's instructions with the following modifications. 0.5 $\mu l/ml$ Brdu was added to the suspension organoid cultures for 1 h at 37°C in the shaking incubator. The cells were harvested with 1 ml Advanced DMEM-F12, spun 5 min at 300$g$ and trypsinised 15 min with TrypLE and DNAseI (20 $\mu g/ml$) in a 37°C water bath. Wash buffer (DMEM +1% FBS) was added to the reaction and the cells were spun at 400$g$ for 5 min. The cells were washed again with 1 ml wash buffer and passed through a 35-$\mu m$ cell strainer. 500,000 cells from each sample were stained with 1:500 Zombie UV (423107; BioLegend) dye in 1× PBS and incubated in darkness for 15 min. 1 ml FACS buffer (1× PBS+1% FBS) was added and cells were spun at 300$g$ for 5 min followed by washes according to the manufacturer's protocol (#370704; BioLegend). 1:100 a-Brdu antibody was added to the FACS buffer (1× PBS+1% FBS) for 20 min at RT in darkness followed by one wash. Cells were spun at 300$g$ for 5 min and incubated with 1:100 7-AAD in FACS buffer for 30 min, before running on a BD LSRFortessa flow cytometer and analysis with BD FACSDiva and FlowJo softwares.

## Data Availability

All RNA-seq data presented here are available at Gene Expression Omnibus, accession number: GSE205753.

## Supplementary Information

## Acknowledgements

We would like to thank members of the Tessarz lab and Achim Tresch for discussions. We are particularly grateful to Niklas Kleinenkuhnen for critical reading of the manuscript. We would like to thank Pauline Camelot for help with setting up some of the organoid cultures and Lena Schumacher for the help with FACS analysis. We are indebted to the FACS and Imaging facility of the MPI for Biology of Ageing and the Imaging Facility of CECAD, University of Cologne for help with imaging. Sequencing was performed at the Sequencing Core Facility of the MPI for Molecular Genetics in Berlin, Germany. This work was funded by the Max Planck Society (to P Tessarz), the BOOST program of the Max Planck Society (to C Nikopoulou) and a Cologne Graduate School for Ageing Research Master fellowship (to K Gehling).

### Author Contributions

K Gehling: formal analysis, investigation, and methodology.
S Parekh: formal analysis.
F Schneider: investigation.
M Kirchner: software and visualization.
V Kondylis: investigation and methodology.
C Nikopoulou: conceptualization, supervision, investigation, methodology, project administration, and writing—original draft, review, and editing.
P Tessarz: conceptualization, supervision, funding acquisition, project administration, and writing—original draft, review, and editing.

## Conflict of Interest Statement

The authors declare that they have no conflict of interest.

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
