## [Reviewer comments · Life Science Alliance]

Life Science Alliance

RNA-seq of single cholangiocyte-derived organoids reveals high organoid-to-organoid variability

Kristin Gehling, Swati Parekh, Farina Schneider, Marcel Kirchner, Vangelis Kondylis, Chrysa Nikopoulou, and Peter Tessarz
DOI: 10.26508/lsa.202101340

Corresponding author(s): Peter Tessarz, Max Planck Institute for Biology of Ageing and Chrysa Nikopoulou

Review Timeline:

Submission Date:	2021-12-16
Editorial Decision:	2022-01-17
Revision Received:	2022-06-08
Editorial Decision:	2022-07-08
Revision Received:	2022-07-14
Accepted:	2022-07-15

Scientific Editor: Novella Guidi

Transaction Report:

January 17, 2022

Re: Life Science Alliance manuscript #LSA-2021-01340

Dr Peter Tessarz
Max Planck Institute for Biology of Ageing
Joseph-Stelzmann-Str. 9b
Cologne, NRW 50931
Germany

Dear Dr. Tessarz,

Thank you for submitting your manuscript entitled "Single organoid RNA-sequencing reveals high organoid-to-organoid variability" to Life Science Alliance. The manuscript was assessed by expert reviewers, whose comments are appended to this letter. We, thus, encourage you to submit a revised version of the manuscript back to LSA that responds to all of the reviewers' points.

Thank you for this interesting contribution to Life Science Alliance. We are looking forward to receiving your revised manuscript.

Sincerely,

B. MANUSCRIPT ORGANIZATION AND FORMATTING:

Reviewer #1 (Comments to the Authors (Required)):

SUMMARY:

The manuscript by Gehling et al provides bulk RNAseq data from murine liver-derived organoids formed in a Matrigel dome vs. in shaking culture to show that the organoids are very different. Bulk RNAseq data was then generated on individual organoids established from 4 different animals and analyzed at different passages. This showed that there were significant gene expression differences between sets of organoid cultures, and the authors concluded this could be attributed to passage number. They also found significant variability in gene expression between individual organoids, which could not be explained by organoid size or cell cycle phase alone, and therefore concluded that this variability could be attributed to microenvironmental differences.

While I believe the RNAseq data generated on individual organoids to showcase the variability between organoids established from genetically identical mice under identical conditions is important and of interest to the scientific community using organoid models, I have significant concerns about how these experiments were designed and executed as outlined below.

MAJOR COMMENTS:

- Page 9: "...even within a single organoid, there can be a high degree of variability as seen in the individual stainings." Please clarify whether this is intended to mean there is a high degree of variability in the staining between individual organoids, or a high degree of variability in the staining of individual cells within a single organoid. If the authors mean the latter, it is not very obvious from Figure 1B that there is a high degree of variability in staining of individual cells. If this is the point the authors are trying to make, please provide clear data to substantiate this point.
- Page 9: "We have additionally tested the reproducibility of organoid generation." Please provide the data/evidence referred to here about the reproducibility of organoid generation.
- The authors refer to Supplementary Tables 1 - 3 but these are missing and could not be reviewed.
- Figure 2C and 2D: legend is unclear as there is no explanation as to how to interpret the difference between what is shown in 2C vs. 2D
- What is the purpose of growing the organoids in a shaking culture (compared to the established Matrigel dome method)? What is the rationale for investigating the gene expression differences between organoids formed in these two methods? If the organoids formed in shaking culture is a new method, there needs to be substantial evidence to demonstrate these organoids are viable, can be propagated, and contain the expected composition of cell types within the organoid by methods such as but not limited to single cell RNA sequencing or immunohistochemistry.
- The authors demonstrate that there is significant technical variability between organoid cultures and recommend that multiple technical replicates need to be performed in organoid experiments. However, they then claim that the biggest gene expression changes between individual organoids was due to how many times the organoids were passaged. However, this claim is made from two replicates at passage 3, and one replicate each at passage 4 and passage 11. The low number of replicates at each passage does not allow for a strong claim that the major difference in gene expression is due to passage number rather than technical variability as the authors themselves demonstrate. Furthermore, to make a strong claim that the gene expression changes are due to passage number, they should derive multiple organoid cultures each from a different animal as they did, but then sample the organoids at different passages from each of these cultures to show there is a similar change in gene expression with passaging that can be seen reproduced between the different cultures.

MINOR COMMENTS:

- Fix the grammatical error on page 5: "For initiating organoid cultured in a dilute suspension culture..."
- Page 10: Was there a specific reason the different pools of murine livers contained different number of animals between pools, or was this simply a consequence of animal availability at the time the study was conducted?
- Please be consistent in how you refer to 'zUMIs' vs. 'ZUMIs' e.g. page 8 vs. page 10
- The acronym DEGs is first defined in the legend for Supplementary Figure 2, when it is first used in Figure 2, and therefore should be first defined in Figure 2.
- Bottom of page 10: "After establishment of organoid culture in domes, cells were subsequently seeded in suspension culture for at least two passages." Please clarify whether this means a suspension of dissociated cells were seeded into suspension culture, or the organoids formed in dome culture were then transferred into suspension culture for an additional two or more passages.

- Figure 3 legend: DEA acronym not defined
- Figure 3D: why is there striking out of the text "by BH procedure"?
- Page 16: "...in which the size in which the size"

Reviewer #2 (Comments to the Authors (Required)):

1. Presented paper addresses important issue of possible gene expression variation in organoid culture. The authors observe differences in obtained organoid cultures and suggest that observed heterogeneity might be resulting from differences of passage number at which the organoids were collected. They also describe heterogeneity between individually grown single organoids regardless of their passage number. Those issues have not been broadly touched before and lay in opposition to most reports suggesting low gene expression variability in organoid cultures and high concordance with material of origin.
2. Variability due to the passage number of pool-derived organoids. The claims states by the authors seem to be valid, as culture conditions were kept identical for all cultures. However, the authors claim that they didn't observe differences in culture even if they were conducted by different person and later on don't provide information if the cultures with differences in expression (likely due to the different passage number) were not conducted by different people. One information lacking from the study would be the comparison of the expression profiles of pool-derived organoids with material of origin and assessment whether the changes in expression grow or are diminished with passage number and therefore culture time. I am, however aware that it would be another lengthy experiment, requiring up to 5 month of time and probably some additional ethical approvals.
 Variability between single organoids. Presented results support the high heterogeneity claim quite well, there are no easily detectable confounding factors present. If available, the photographs or other data describing the cultures in more detail would be welcome, as the authors suggest "that intra-well conditions might be a strong driver of culture variability". Interesting addition to the study would be to test whether specific differentiating culture conditions alleviate observed expression heterogeneity as the differences are claimed to come mostly from varying differentiation state of the organoids.
 Assessment of the cell cycle phase. The authors used transcriptome based approach to asses the cell cycle phase the organoids were in. Maybe additional analysis with flow cytometry could confirm their findings, but again to obtain fresh material it would require another organoid culture performed.
3. The manuscript is well written and only minor corrections of text repetitions are required.

Reviewer #3 (Comments to the Authors (Required)):

In this study, Gehling et al performed a transcriptomic analysis of liver organoids derived from different mice to address heterogeneity between mice, organoid propagation method (fixed vs shaking), and single organoids in culture. The authors show through various gene expression analyses the variability depending on the method of liver organoid propagation (prepared in solid matrigel versus transfer to a shaking protocol), the mouse used, and presents some limited data suggesting passage number affects gene expression. In particular, authors found 1) organoids in fixed matrigel culture express higher levels of mature liver markers and organoids propagated into a shaking culture express more genes suggesting elevated proliferation, 2) single organoids from the same culture have different gene expression patterns, and 3) passage level appears to affect organoid maturity.

In general, there is a tendency to over-interpretation of findings based on small number of biological replicates for claims on passage number and mouse differences. Additionally, deficiencies in experimental design limit the robustness of the results. Though the rationale of studying factors that affect liver organoid heterogeneity is important, the data as presented do not sufficiently support all the claims of the paper. With the experimental design and data currently in hand, this study could benefit from tempering the claims (detailed below), improved clarity in the methods, and re-interpreting or re-enforcing some of the results (also detailed below).

General points:

-The article makes broad claims about organoid culture in general, but only presents data on liver organoids. The claims are not generalizable to all organoid culture as methods used to prepare different and maintain different organ/tissue types vary widely and likely have different results. The title and abstract do not mention liver at all, and need to reflect the tissue studied, and type of organoid (hep, chol..). Furthermore, the abstract makes claims about batch-to-batch variability between researchers that are not backed up by any provided data. This may be the case, but could be included in discussion as a data not shown even, but if it's in the abstract it needs to be backed up by data.

-There is confusion in the study design with regard to selection of culture passages to study. Why 2 at passage 3, 1 at passage 4, and 1 at passage 11. Unfortunately this design makes it difficult to interpret the robustness of the findings as it is essentially making broad claims on maturity of organoids relating to passage number with only an n=1 at this late passage stage.

-Were the mice used littermates, co-housed, all collected at the same time? Age and sex likely contribute to disparate findings,

thus these need to be reported especially for figure 2 where they could confound the data with regard to what is driving gene expression differences.

-The strongest data is in figure 4 looking at individual organoid expression in single isolated organoids and interesting analysis with Cyclone. Appropriately the authors did a nice analysis comparing to organoid area and didn't find any significant trends. This heterogeneity could well be due to a number of things like epigenetics of the particular stem cell/organoid forming unit, other cells within the local vicinity, exposure to nutrients in the initial fixed matrigel culture (i.e. edge vs middle). It is not within the scope of this paper to determine these things, but their mention is important. This is lightly touched on in the abstract as 'microenvironment factors' and briefly as 'cell to cell contacts' in discussion, but this needs to be explored further as a possible explanation in results or discussion.

-Supplemental tables referenced in the text are not provided. Some proofreading required for typos, missed parentheses etc. Introduction discusses Fah mice should briefly mention why these mice are a relevant model to the point being made.

-As this paper relies heavily on method of culture preparation, it needs to have a solid and rigorous methods section. Same litters used, sexes of mice, age of mice used for Pools A, B, and C. Sets 1-4 are defined in text, but include in methods. The modifications of the manufacturer's protocol need to be clearly stated so they can be replicated and assessed if they contribute to differences observed. Describe the method of passage, syringe, pipet-mediated shearing, etc? What type of matrigel used (catalog numbers, GFR, etc..). Method of passage from suspension culture. What does "organized in the best manner" mean for the RNA collection. Catalog numbers are lacking on many reagents used (i.e. what type of PBS? Additives to DMEM/F12?). What antibodies were used for IHC staining, times, & all concentrations, this is not mentioned in methods. Versions of FIJI and ImageJ and if any plugins were required for analysis. Read depth of RNAseq.

Specific critiques:

Figure 1

-Please include rationale for looking at these specific markers by IHC. I'm unable to see differences between Hnf4 and Ki67 staining within this culture prep as is stated in the results. Perhaps a higher resolution inset and quantification could reflect this. The rationale that there are differences in expression is correct based on the authors figure 3 data, but this figure does not illustrate this to me. Additionally, type of organoid needs to be mentioned here (liver and sub-type what were they generated from).

-In the figure legend, what does young organoid indicate.

Figure 2

-Needs to include biological replicates used to generate figure in the legend. DEG is presented without definition, please include in legend.

Figure 3

-Passage number contributing strongest to gene expression changes is difficult to accept this claim because there is only n=1 for Set1/passage 11 which the results is based on. These results could be due to mouse (ie what if one was sick?) or litter, but it is not indicated if all mice were from the same litters for these experiments. Despite this, this claim would need more biological replicates to establish this conclusion. E and F are difficult to read as printed. DEA not defined in legend.

-Please clarify singularity per well in the results text.

Figure 3

-Include rationale for looking at Wnt

Overall a re-focusing on the aspects of this study that are robust would greatly improve this paper in the absence of additional experimentation.

We would like to thank the reviewers for their insightful and constructive criticism. We have addressed all the comments, particularly focused and clarified the text and added more experimental data. Text changes are highlighted in yellow. Please find more information in our response letter. We hope that you'll find the revised version of the manuscript now acceptable for publication.

Reviewer #1 (Comments to the Authors (Required)):

SUMMARY:

The manuscript by Gehling et al provides bulk RNAseq data from murine liver-derived organoids formed in a Matrigel dome vs. in shaking culture to show that the organoids are very different. Bulk RNAseq data was then generated on individual organoids established from 4 different animals and analyzed at different passages. This showed that there were significant gene expression differences between sets of organoid cultures, and the authors concluded this could be attributed to passage number. They also found significant variability in gene expression between individual organoids, which could not be explained by organoid size or cell cycle phase alone, and therefore concluded that this variability could be attributed to microenvironmental differences.

While I believe the RNAseq data generated on individual organoids to showcase the variability between organoids established from genetically identical mice under identical conditions is important and of interest to the scientific community using organoid models, I have significant concerns about how these experiments were designed and executed as outlined below.

We thank the reviewer for the clear and constructive criticism that we answer to, below.

MAJOR COMMENTS:

- Page 9: "...even within a single organoid, there can be a high degree of variability as seen in the individual stainings." Please clarify whether this is intended to mean there is a high degree of variability in the staining between individual organoids, or a high degree of variability in the staining of individual cells within a single organoid. If the authors mean the latter, it is not very obvious from Figure 1B that there is a high degree of variability in staining of individual cells. If this is the point the authors are trying to make, please provide clear data to substantiate this point.

We are sorry that we were not clear enough on this part. We wanted to showcase that cells within an organoid, individual organoids (and not cells within one organoid) differ substantially. We have now amended the text (lines 97-99) and provided new pictures with more organoids, combined with a zoom-in to make this point more obvious visually (Figure 1B-C, Supplementary Figure 1.

- Page 9: "We have additionally tested the reproducibility of organoid generation." Please provide the data/evidence referred to here about the reproducibility of organoid generation.

We agree with the reviewer that this sentence feels out of place at this point. We have deleted this sentence here. We have now added a sentence to the discussion (line 234-240) regarding the issue on batch-to-batch reproducibility.

- The authors refer to Supplementary Tables 1 - 3 but these are missing and could not be reviewed.

We apologize for this mistake. The manuscript was transferred from bioRxiv and we did not find a way to attach the supplementary tables. Please find them now as part of the new submission.

- Figure 2C and 2D: legend is unclear as there is no explanation as to how to interpret the difference between what is shown in 2C vs. 2D

We apologise and have added the respective information.

- What is the purpose of growing the organoids in a shaking culture (compared to the established Matrigel dome method)? What is the rationale for investigating the gene expression differences between organoids formed in these two methods? If the organoids formed in shaking culture is a new method, there needs to be substantial evidence to demonstrate these organoids are viable, can be propagated, and contain the expected composition of cell types within the organoid by methods such as but not limited to single cell RNA sequencing or immunohistochemistry.

Growth of liver-derived organoids in shaking culture is not done frequently, but is suggested by Stem Cell Technologies to be used in case one is interested in generating high cell numbers (<https://www.stemcell.com/products/hepaticult-organoid-growth-medium-mouse.html#section-product-use>) as a standard way of expanding organoids. The company also provides an initial quality assessment and comparison between dome and suspension grown organoids. This approach has also been used in human liver-derived organoids to generate large cell numbers of Lgr5 positive cells (Schneeberger et al., 2019; <https://doi.org/10.1002/hep.31037>). Our rationale for using organoids in shaking cultures was the fact that – at least in our hands – isolation of intact organoids from dome cultures was very difficult, especially the isolation of intact organoids for subsequent microscopy that we wanted to do to be able to correlate any gene expression change with the observed size differences. We performed a general RNA-seq to compare dome-vs-suspension organoids in our hands. As we think that this is a valuable data set on its own and provides the reader with the possibility to compare our results with previous reports, we decided to also include this data in the manuscript. We have now made these points clearer in the revised manuscript (lines 106-110, 132-134, 288-300).

- The authors demonstrate that there is significant technical variability between organoid cultures and recommend that multiple technical replicates need to be performed in organoid experiments. However, they then claim that the biggest gene expression changes between individual organoids was due to how many times the organoids were passaged. However, this claim is made from two replicates at passage 3, and one replicate each at passage 4 and passage 11. The low number of replicates at each passage does not allow for a strong claim that the major difference in gene expression is due to passage number rather than technical variability as the authors themselves demonstrate. Furthermore, to make a strong claim that the gene expression changes are due to passage number, they should derive multiple organoid cultures each from a different animal as they did, but then sample the

organoids at different passages from each of these cultures to show there is a similar change in gene expression with passaging that can be seen reproduced between the different cultures.

We agree with the reviewer and have now performed an RNA-seq experiment, in which we followed individual (n=3) organoid batches at low and high passages. This result is included as new Figure 4 and supports the previous observation that passage number is the main driver for expression change differences.

MINOR COMMENTS:

- Fix the grammatical error on page 5: "For initiating organoid cultured in a dilute suspension culture..."

fixed

- Page 10: Was there a specific reason the different pools of murine livers contained different number of animals between pools, or was this simply a consequence of animal availability at the time the study was conducted?

We deleted this information from the result section and moved it to Materials and Methods, where we also clarify that this was due to animal availability at time of dissection.

- Please be consistent in how you refer to 'zUMIs' vs. 'ZUMIs' e.g. page 8 vs. page 10

All changed to zUMIs.

- The acronym DEGs is first defined in the legend for Supplementary Figure 2, when it is first used in Figure 2, and therefore should be first defined in Figure 2.

We have now defined the acronym in Figure 2.

- Bottom of page 10: "After establishment of organoid culture in domes, cells were subsequently seeded in suspension culture for at least two passages." Please clarify whether this means a suspension of dissociated cells were seeded into suspension culture, or the organoids formed in dome culture were then transferred into suspension culture for an additional two or more passages.

We apologise that we were not clear here. During passaging, the organoids are broken down into fragments, which are then plated again into either dome or suspension culture (this is the standard way of passaging). We have tried to make this clearer in the revised version of the manuscript (lines 288-290).

- Figure 3 legend: DEA acronym not defined

Now defined.

- Figure 3D: why is there striking out of the text "by BH procedure"?

This was a left-over from the editing of the manuscript. This has been removed now.

- Page 16: "...in which the size in which the size"

corrected

Reviewer #2 (Comments to the Authors (Required)):

1. Presented paper addresses important issue of possible gene expression variation in organoid culture. The authors observe differences in obtained organoid cultures and suggest that observed heterogeneity might be resulting from differences of passage number at which the organoids were collected. They also describe heterogeneity between individually grown single organoids regardless of their passage number. Those issues have not been broadly touched before and lay in opposition to most reports suggesting low gene expression variability in organoid cultures and high concordance with material of origin.

This point is very important. There are two things to consider. First, there is indeed low gene expression variability between different batches of organoids and we can confirm this using bulk RNA-seq (see Figure 2B and new Figure 4A). However, when one analyzes single organoids, it is apparent that there is heterogeneity in one single tissue-culture well. These two concepts – bulk vs. single organoid is very important to bear in mind when designing experiments.

2. Variability due to the passage number of pool-derived organoids. The claims states by the authors seem to be valid, as culture conditions were kept identical for all cultures. However, the authors claim that they didn't observe differences in culture even if they were conducted by different person and later on don't provide information if the cultures with differences in expression (likely due to the different passage number) were not conducted by different people. One information lacking from the study would be the comparison of the expression profiles of pool-derived organoids with material of origin and assessment whether the changes in expression grow or are diminished with passage number and therefore culture time. I am, however aware that it would be another lengthy experiment, requiring up to 5 month of time and probably some additional ethical approvals.

We agree with the reviewer and have now performed an RNA-seq experiment, in which we followed individual (n=3) organoid batches over several passages. This result is included as new Figure 4 and supports the previous observation that passage number is the main driver for expression change differences. We have not included a comparison between tissue of origin and organoid. We believe that this comparison is not telling, particularly in liver-derived organoids. The tissue would be dominated by a hepatic signature, whereas the organoid would reflect more progenitor-like states.

Variability between single organoids. Presented results support the high heterogeneity claim quite well, there are no easily detectable confounding factors present. If available, the photographs or other data describing the cultures in more detail would be welcome, as the authors suggest "that intra-well conditions might be a strong driver of culture variability".

We agree that this would be a nice addition to the manuscript to give the reader an impression how different the microenvironment of each organoid is like. We have included these data as new Supplementary Figure 5.

Interesting addition to the study would be to test whether specific differentiating culture conditions alleviate observed expression heterogeneity as the differences are claimed to come mostly from varying differentiation state of the organoids.

We agree with the reviewer that this would be an interesting point to investigate, but think that this is outside the scope for this study.

Assessment of the cell cycle phase. The authors used transcriptome based approach to assess the cell cycle phase the organoids were in. Maybe additional analysis with flow cytometry could confirm their findings, but again to obtain fresh material it would require another organoid culture performed.

As suggested, we performed a cell cycle analysis by flow cytometry. The results are shown as new Supplementary Figure 4C-D and confirm the Cyclone analysis.

3. The manuscript is well written and only minor corrections of text repetitions are required.

We carefully proofread the current manuscript.

Reviewer #3 (Comments to the Authors (Required)):

In this study, Gehling et al performed a transcriptomic analysis of liver organoids derived from different mice to address heterogeneity between mice, organoid propagation method (fixed vs shaking), and single organoids in culture. The authors show through various gene expression analyses the variability depending on the method of liver organoid propagation (prepared in solid matrigel versus transfer to a shaking protocol), the mouse used, and presents some limited data suggesting passage number affects gene expression. In particular, authors found 1) organoids in fixed matrigel culture express higher levels of mature liver markers and organoids propagated into a shaking culture express more genes suggesting elevated proliferation, 2) single organoids from the same culture have different gene expression patterns, and 3) passage level appears to affect organoid maturity.

In general, there is a tendency to over-interpretation of findings based on small number of biological replicates for claims on passage number and mouse differences. Additionally, deficiencies in experimental design limit the robustness of the results. Though the rationale of studying factors that affect liver organoid heterogeneity is important, the data as presented do not sufficiently support all the claims of the paper. With the experimental design and data currently in hand, this study could benefit from tempering the claims (detailed below), improved clarity in the methods, and re-interpreting or re-enforcing some of the results (also detailed below).

We thank the reviewer for the clear and constructive criticism. We have tried and add more clarity to the text, provide more details on the experimental design and added new data to support our claims.

General points:

-The article makes broad claims about organoid culture in general, but only presents data on liver organoids. The claims are not generalizable to all organoid culture as methods used to prepare different and maintain different organ/tissue types vary widely and likely have different results. The title and abstract do not mention liver at all, and need to reflect the tissue studied, and type of organoid (hep, chol..). Furthermore, the abstract makes claims about batch-to-batch variability between researchers that are not backed up by any provided data. This may be the case, but could be included in discussion as a data not shown even, but if it's in the abstract it needs to be backed up by data.

We fully agree with the reviewer and have made the corresponding changes to the manuscript: i) we now mention that we designed the study based on liver-derived organoids in the abstract (now called intrahepatic cholangiocyte organoids according to the guidelines published in (Marsee et al, 2021), the introduction (line 59) and results (line 78) sections; ii) we removed the point on the researcher's impact on batch-to-batch reproducibility from the abstract and results sections and added a sentence to the discussion (line 237) to tone down this aspect.

-There is confusion in the study design with regard to selection of culture passages to study. Why 2 at passage 3, 1 at passage 4, and 1 at passage 11. Unfortunately this design makes it difficult to interpret the robustness of the findings as it is essentially making broad claims on maturity of organoids relating to passage number with only an n=1 at this late passage stage.

We agree with the reviewer. The study design never had in mind to do a cross-passage comparison and we have re-written this section in the result section to avoid any confusion about this point (line 148-150). This finding came only after the analysis of the single-organoid data. To support the finding of a strong passage effect, we have now performed an RNA-seq experiment, in which we followed individual (n=3) organoid batches over several passages. This result is included as new Figure 4 and supports the previous observation that passage number is the main driver for expression change differences.

-Were the mice used littermates, co-housed, all collected at the same time? Age and sex likely contribute to disparate findings, thus these need to be reported especially for figure 2 where they could confound the data with regard to what is driving gene expression differences.

We have added the details to the used mice as part of the Materials & Methods section. Not all mice used as donors were litter mates, nor collected at the same time. However, they were co-housed and from the same sex and age.

-The strongest data is in figure 4 looking at individual organoid expression in single isolated organoids and interesting analysis with Cyclone. Appropriately the authors did a nice analysis comparing to organoid area and didn't find any significant trends. This

heterogeneity could well be due to a number of things like epigenetics of the particular stem cell/organoid forming unit, other cells within the local vicinity, exposure to nutrients in the initial fixed matrigel culture (i.e. edge vs middle). It is not within the scope of this paper to determine these things, but their mention is important. This is lightly touched on in the abstract as 'microenvironment factors' and briefly as 'cell to cell contacts' in discussion, but this needs to be explored further as a possible explanation in results or discussion.

We thank the reviewer for pointing this out as this is an important aspect. We have expanded the discussion section further to include a more thorough discourse of potential drivers of heterogeneity (lines 246-247). In addition, and to give the reader a better idea about the culture conditions, we have now added photographs of shaking cultures to showcase the difference in the direct environments of different organoids (new Supplementary Figure 5).

-Supplemental tables referenced in the text are not provided. Some proofreading required for typos, missed parentheses etc. Introduction discusses Fah mice should briefly mention why these mice are a relevant model to the point being made.

We agree that Fah^{-/-} mice did not add any significant insight into the introduction and we have removed this part from the introduction. We apologise for the absence of supplementary tables. The manuscript was transferred from bioRxiv and we did not find a way to attach the supplementary tables. Please find them now as part of the new submission. We carefully proofread the manuscript for typos.

-As this paper relies heavily on method of culture preparation, it needs to have a solid and rigorous methods section. Same litters used, sexes of mice, age of mice used for Pools A, B, and C. Sets 1-4 are defined in text, but include in methods.

We agree and have now given all the details in the Materials and Methods section (line 263).

The modifications of the manufacturer's protocol need to be clearly stated so they can be replicated and assessed if they contribute to differences observed. Describe the method of passage, syringe, pipet-mediated shearing, etc? What type of matrigel used (catalog numbers, GFR, etc..). Method of passage from suspension culture. What does "organized in the best manner" mean for the RNA collection. Catalog numbers are lacking on many reagents used (i.e. what type of PBS? Additives to DMEM/F12?). What antibodies were used for IHC staining, times, & all concentrations, this is not mentioned in methods. Versions of FIJI and ImageJ and if any plugins were required for analysis. Read depth of RNAseq.

We have added all relevant information to the revised manuscript.

Specific critiques:

Figure 1

-Please include rationale for looking at these specific markers by IHC. I'm unable to see differences between Hnf4 and Ki67 staining within this culture prep as is stated in the results. Perhaps a higher resolution inset and quantification could reflect this. The rationale that there are differences in expression is correct based on the authors figure 3 data, but this

figure does not illustrate this to me. Additionally, type of organoid needs to be mentioned here (liver and sub-type what were they generated from).

We have now provided new pictures with more organoids, combined with a zoom-in to make this point more obvious visually and added information to figure legend about organoid type.

-In the figure legend, what does young organoid indicate.

Clarified in the text.

Figure 2

-Needs to include biological replicates used to generate figure in the legend. DEG is presented without definition, please include in legend.

Information added.

Figure 3

-Passage number contributing strongest to gene expression changes is difficult to accept this claim because there is only n=1 for Set1/passage 11 which the results are based on. These results could be due to mouse (ie what if one was sick?) or litter, but it is not indicated if all mice were from the same litters for these experiments. Despite this, this claim would need more biological replicates to establish this conclusion. E and F are difficult to read as printed. DEA not defined in legend.

For replicates, please see above. Added paragraph and new figure with RNA-seq of the same batch at two different passage numbers. DEA now defined. We have tried to increase the resolution of E and F.

-Please clarify singularity per well in the results text.

We apologise if this term was not clear. It should refer to the fact that only one organoid was present in the well before RNA-extraction. We have made this now clearer in the revised version of the manuscript.

Figure 3

-Include rationale for looking at Wnt

We assume that the reviewer is referring to Figure 4. Wnt signalling is an essential contributor of long-term organoid expansion. We have added this information to the text (line 198).

Overall a re-focusing on the aspects of this study that are robust would greatly improve this paper in the absence of additional experimentation.

We hope that the changes made have improved the manuscript to support publication.

July 8, 2022

RE: Life Science Alliance Manuscript #LSA-2021-01340R

Dr. Peter Tessarz
Max Planck Institute for Biology of Ageing
Joseph-Stelzmann-Str. 9b
Cologne, NRW 50931
Germany

Dear Dr. Tessarz,

Thank you for submitting your revised manuscript entitled "RNA-seq of single cholangiocyte-derived organoids reveals high organoid-to-organoid variability". We would be happy to publish your paper in Life Science Alliance pending final revisions necessary to meet our formatting guidelines.

- please address the final Reviewer #2 and #3 concerns via discussion. No additional experiments are required. Please address their concerns in a point-by-point letter format
- please add the ORCID ID for the secondary corresponding author-you should have received instructions on how to do so

A. FINAL FILES:

- An editable version of the final text (.DOC or .DOCX) is needed for copyediting (no PDFs).

- High-resolution figure, supplementary figure and video files uploaded as individual files: See our detailed guidelines for preparing your production-ready images, <https://www.life-science-alliance.org/authors>

- Summary blurb (enter in submission system): A short text summarizing in a single sentence the study (max. 200 characters including spaces). This text is used in conjunction with the titles of papers, hence should be informative and complementary to the title. It should describe the context and significance of the findings for a general readership; it should be written in the present tense and refer to the work in the third person. Author names should not be mentioned.

B. MANUSCRIPT ORGANIZATION AND FORMATTING:

Sincerely,

Reviewer #1 (Comments to the Authors (Required)):

I am satisfied that the authors have addressed my initial comments. I believe the addition of new data and figures as requested has improved the manuscript, and the results will be of interest and importance to the scientific community using organoid models to study disease.

Reviewer #2 (Comments to the Authors (Required)):

1. Presented paper addresses important issue of possible gene expression variation in organoid culture. The authors observe differences in obtained organoid cultures and suggest that observed heterogeneity might be resulting from differences of passage number at which the organoids were collected. They also describe heterogeneity between individually grown single organoids regardless of their passage number. Those issues have not been broadly touched before and lay in opposition to most reports suggesting low gene expression variability in organoid cultures and high concordance with material of origin.

This point is very important. There are two things to consider. First, there is indeed low gene expression variability between different batches of organoids and we can confirm this using bulk RNA-seq (see Figure 2B and new Figure 4A). However, when one analyzes single organoids, it is apparent that there is heterogeneity in one single tissue-culture well. These two concepts - bulk vs. single organoid is very important to bear in mind when designing experiments.

I have a problem with the definition of bulk/single organoid in this paragraph. If the organoids were cultured as singletons, would the single well not be considered a batch? Just a food for thought.

But what is important here and not explored in presented paper is the interaction between organoids within single well and the microenvironment that they create (in batch culture) and lack thereof in single organoids.

The authors acknowledge "that intra-well conditions might be a strong driver of culture variability", but do not analyze them further.

2. R2: Variability due to the passage number of pool-derived organoids. The claims states by the authors seem to be valid, as culture conditions were kept identical for all cultures. However, the authors claim that they didn't observe differences in culture even if they were conducted by different person and later on don't provide information if the cultures with differences in expression (likely due to the different passage number) were not conducted by different people. One information lacking from the study would be the comparison of the expression profiles of pool-derived organoids with material of origin and assessment whether the changes in expression grow or are diminished with passage number and therefore culture time. I am, however aware that it would be another lengthy experiment, requiring up to 5 month of time and probably some additional ethical approvals.

A: We agree with the reviewer and have now performed an RNA-seq experiment, in which we followed individual (n=3) organoid batches over several passages. This result is included as new Figure 4 and supports the previous observation that passage number is the main driver for expression change differences.

R2: The main problem here is that each batch was analyzed at a different passage and not used to generate material at different passage numbers and then analyzed for inter- and intrapool variability in gene expression. As stated: "Organoids from Set 1 were analysed after passage 11 and displayed a similar morphology within the set regarding the evenness of the lumen compared to all other sets. Sets 2, 3 were harvested at passage three and Set 4 at passage four. Organoid cultures for the heterogeneity experiment (Set 1 - Set 4) were established from the liver of only one mouse for each set. No physiological abnormalities, like tumours, were observed during dissection of the mice."

The analysis of gene expression in organoids from the same set, but different passage would indicate if the observed differences in fact arose over time and are passage bound or if they were present from the initiation of the culture.

Also, please clarify the nomenclature of the batch/set etc, because they are quite confounding.

A: We have not included a comparison between tissue of origin and

organoid. We believe that this comparison is not telling, particularly in liver-derived organoids. The tissue would be dominated by a hepatic signature, whereas the organoid would reflect more progenitor-like states.

R2: The differences in cellular composition as well as gene expression patterns are crucial for the fate of the resulting organoid, therefore I believe that inclusion of such data is a necessity.

R2: Variability between single organoids. Presented results support the high heterogeneity claim quite well, there are no easily detectable confounding factors present. If available, the photographs or other data describing the cultures in more detail would be welcome, as the authors suggest "that intra-well conditions might be a strong driver of culture variability".

A: We agree that this would be a nice addition to the manuscript to give the reader an impression how different the microenvironment of each organoid is like. We have included these data as new Supplementary Figure 5.

R2: Referred to above.

R2: Interesting addition to the study would be to test whether specific differentiating culture conditions alleviate observed expression heterogeneity as the differences are claimed to come mostly from varying differentiation state of the organoids.

A: We agree with the reviewer that this would be an interesting point to investigate, but think that this is outside the scope for this study.

R2: When presenting a problem it would be appreciated if a solution to that problem was also presented, but I accept the justification of the authors on this matter.

R2: Assessment of the cell cycle phase. The authors used transcriptome based approach to assess the cell cycle phase the organoids were in. Maybe additional analysis with flow cytometry could confirm their findings, but again to obtain fresh material it would require another organoid culture performed.

A: As suggested, we performed a cell cycle analysis by flow cytometry. The results are shown as new Supplementary Figure 4C-D and confirm the Cyclone analysis.

R2: OK

3. The manuscript is well written and only minor corrections of text repetitions are required.

A: We carefully proofread the current manuscript.

R2: OK

Reviewer #3 (Comments to the Authors (Required)):

This revised manuscript by Gehling et al has tempered their conclusions to their data presented on liver organoids. Authors have added additional RNA sequencing to reinforce the finding that organoid passage number contributes to gene expression changes that, importantly, looks at organoids generated from the same batch across passage. Though the study would benefit from higher powered design and controls, the data shown suggest passage number affects gene expression and shows variability in individual liver organoids (Fig 5). I still cannot appreciate the variability in HNF4A and SOX9 in Fig 1 in the far left organoid versus others for both stains as this could simply reflect sectioning artifacts of embedded organoids. This presentation is not convincing. Is morphology different between these organoids? Are some a single cell thickness layer versus the far left organoid which appears to be either multiple cells thick, or cut at a sheared angle. The differences in Ki67 and size are apparent. Perhaps images similar to Fig 3A demonstrating this altered morphology if it exists would support the point of the figure, another quantification or set of images, or better explanation.

We would like to thank the reviewers again for their insightful comments and their support in publishing our manuscript. Please find the point-by-point responses below. For ease of re-review we have highlighted the edited text in green.

Reviewer #1

I am satisfied that the authors have addressed my initial comments. I believe the addition of new data and figures as requested has improved the manuscript, and the results will be of interest and importance to the scientific community using organoid models to study disease.

We would like to thank the reviewer for highlighting the significance of our study and supporting publication.

Reviewer #2

1. Presented paper addresses important issue of possible gene expression variation in organoid culture. The authors observe differences in obtained organoid cultures and suggest that observed heterogeneity might be resulting from differences of passage number at which the organoids were collected. They also describe heterogeneity between individually grown single organoids regardless of their passage number. Those issues have not been broadly touched before and lay in opposition to most reports suggesting low gene expression variability in organoid cultures and high concordance with material of origin.

Initial Response: This point is very important. There are two things to consider. First, there is indeed low gene expression variability between different batches of organoids and we can confirm this using bulk RNA-seq (see Figure 2B and new Figure 4A). However, when one analyzes single organoids, it is apparent that there is heterogeneity in one single tissue-culture well. These two concepts - bulk vs. single organoid is very important to bear in mind when designing experiments.

I have a problem with the definition of bulk/single organoid in this paragraph. If the organoids were cultured as singletons, would the single well not be considered a batch? Just a food for thought. But what is important here and not explored in presented paper is the interaction between organoids within single well and the microenvironment that they create (in batch culture) and lack thereof in single organoids.

The authors acknowledge "that intra-well conditions might be a strong driver of culture variability", but do not analyze them further.

We agree with the reviewer that the interaction between the organoids in a single well and the microenvironment can be a significant source of variability. We also apologize for not being clear in the manuscript. We did not grow single organoids in wells, but only isolated them from a well full of organoids for imaging and subsequent lysis and mRNA isolation. Thus, the microenvironment was always strongly influenced by the presence of other organoids within the well.

2. R2: Variability due to the passage number of pool-derived organoids. The claims states by the authors seem to be valid, as culture conditions were kept identical for all cultures. However, the authors claim that they didn't observe differences in culture even if they were conducted by different person and later on don't provide information if the cultures with differences in expression (likely due to the different passage number) were not conducted by different people. One information lacking from the study would be the comparison of the expression profiles of pool-derived organoids with material of origin and assessment whether the changes in expression grow or are diminished with passage number and therefore culture time. I am, however aware that it would be another lengthy experiment, requiring up to 5 month of time and probably some additional ethical approvals.

Initial Response: We agree with the reviewer and have now performed an RNAseq experiment, in which we followed individual (n=3) organoid batches over several passages. This result is included as new Figure 4 and supports the previous observation that passage number is the main driver for expression change differences.

R2: The main problem here is that each batch was analyzed at a different passage and not used to generate material at different passage numbers and then analyzed for inter- and intrapool variability in gene expression. As stated:

"Organoids from Set 1 were analysed after passage 11 and displayed a similar morphology within the set regarding the evenness of the lumen compared to all other sets. Sets 2, 3 were harvested at passage three and Set 4 at passage four. Organoid cultures for the heterogeneity experiment (Set 1 - Set 4) were established from the liver of only one mouse for each set. No physiological abnormalities, like tumours, were observed during dissection of the mice."

The analysis of gene expression in organoids from the same set, but different passage would indicate if the observed differences in fact aroused over time and are passage bound or if they were present from the initiation of the culture.

Also, please clarify the nomenclature of the batch/set etc, because they are quite confounding.

By analyzing each set at a different passage we aimed to address the question whether passaging of the organoids is a confounding factor for differences in gene expression. We added a heatmap (Figure 4B) which shows that the different sets do not show profound changes in gene expression between each other, but there are changes arising from prolonged culturing. We agree with the reviewer that the nomenclature might be confusing, therefore we added an additional table (Table S5, lines 260-261) with information about all organoid batches/sets that were used in this study.

Initial Response: We have not included a comparison between tissue of origin and organoid. We believe that this comparison is not telling, particularly in liver-derived organoids. The tissue would be dominated by a hepatic signature, whereas the organoid would reflect more progenitor-like states.

R2: The differences in cellular composition as well as gene expression patterns are crucial for the fate of the resulting organoid, therefore I believe that inclusion of such data a necessity.

We did not aim to include a study comparing the organoids with the tissue of origin because this was done when the protocol was originally established [1], showing that the organoid cells are much different from those of the tissue of origin. The protocol for establishing intrahepatic

cholangiocyte organoid cultures is well established [2,3] and since we did not observe any differences in the different batches of organoids (Figure 2B, Figure 4B), which would have hinted to variations in animal tissue material, we focused rather on the organoid culture conditions and heterogeneity in gene expression which is caused by the microenvironment within the organoid culture.

R2: Variability between single organoids. Presented results support the high heterogeneity claim quite well, there are no easily detectable confounding factors present. If available, the photographs or other data describing the cultures in more detail would be welcome, as the authors suggest "that intra-well conditions might be a strong driver of culture variability".

Initial Response: We agree that this would be a nice addition to the manuscript to give the reader an impression how different the microenvironment of each organoid is like. We have included these data as new Supplementary Figure 5.

R2: Referred to above.

R2: Interesting addition to the study would be to test whether specific differentiating culture conditions alleviate observed expression heterogeneity as the differences are claimed to come mostly from varying differentiation state of the organoids.

Initial Response: We agree with the reviewer that this would be an interesting point to investigate, but think that this is outside the scope for this study.

R2: When presenting a problem it would be appreciated if a solution to that problem was also presented, but I accept the justification of the authors on this matter.

We thank the reviewer for agreeing that such an experiment would be beyond the scope of this study.

R2: Assessment of the cell cycle phase. The authors used transcriptome based approach to assess the cell cycle phase the organoids were in. Maybe additional analysis with flow cytometry could confirm their findings, but again to obtain fresh material it would require another organoid culture performed.

Initial Response: As suggested, we performed a cell cycle analysis by flow cytometry. The results are shown as new Supplementary Figure 4C-D and confirm the Cyclone analysis.

R2: OK

3. The manuscript is well written and only minor corrections of text repetitions are required.

Initial Response: We carefully proofread the current manuscript.

R2: OK

Reviewer #3

This revised manuscript by Gehling et al has tempered their conclusions to their data presented on liver organoids. Authors have added additional RNA sequencing to reinforce the finding that organoid passage number contributes to gene expression changes that, importantly, looks at organoids generated from the same batch across passage. Though the study would benefit from higher powered design and controls, the data shown suggest passage number affects gene expression and shows variability in individual liver organoids (Fig 5). I still cannot appreciate the variability in HNF4A and SOX9 in Fig 1 in the far left organoid versus others for both stains as this could simply reflect sectioning artifacts of embedded organoids. This presentation is not convincing. Is morphology different between these organoids? Are some a single cell thickness layer versus the far left organoid which appears to be either multiple cells thick, or cut at a sheared angle. The differences in Ki67 and size are apparent. Perhaps images similar to Fig 3A demonstrating this altered morphology if it exists would support the point of the figure, another quantification or set of images, or better explanation.

We agree with the reviewer that the differences are not so profound for HNF4 and SOX9. Although we cannot exclude the possibility of differences in embedding, there are indeed differences in the morphology of the epithelium and differences in the number of positive cells between the different organoids. Therefore we re-wrote this section (line 97-101) to be more clear and precise.

References

1. Huch M, Gehart H, van Boxtel R, et al. Long-term culture of genome-stable bipotent stem cells from adult human liver. *Cell* 2015; 160:299–312
2. Huch M, Dorrell C, Boj SF, et al. In vitro expansion of single Lgr5+ liver stem cells induced by Wnt-driven regeneration. *Nature* 2013; 494:247–250
3. Broutier L, Andersson-Rolf A, Hindley CJ, et al. Culture and establishment of self-renewing human and mouse adult liver and pancreas 3D organoids and their genetic manipulation. *Nat. Protoc.* 2016; 11:1724–1743

July 15, 2022

RE: Life Science Alliance Manuscript #LSA-2021-01340RR

Dr. Peter Tessarz
Max Planck Institute for Biology of Ageing
Joseph-Stelzmann-Str. 9b
Cologne, NRW 50931
Germany

Dear Dr. Tessarz,

Thank you for submitting your Research Article entitled "RNA-seq of single cholangiocyte-derived organoids reveals high organoid-to-organoid variability". It is a pleasure to let you know that your manuscript is now accepted for publication in Life Science Alliance. Congratulations on this interesting work.

DISTRIBUTION OF MATERIALS:

Again, congratulations on a very nice paper. I hope you found the review process to be constructive and are pleased with how the manuscript was handled editorially. We look forward to future exciting submissions from your lab.

Sincerely,
